# Abnormal Micronutrient Intake Is Associated with the Risk of Periodontitis: A Dose–response Association Study Based on NHANES 2009–2014

**DOI:** 10.3390/nu14122466

**Published:** 2022-06-14

**Authors:** Weiqi Li, Qianhui Shang, Dan Yang, Jiakuan Peng, Hang Zhao, Hao Xu, Qianming Chen

**Affiliations:** State Key Laboratory of Oral Diseases, National Clinical Research Center for Oral Diseases, Chinese Academy of Medical Sciences Research Unit of Oral Carcinogenesis and Management, West China Hospital of Stomatology, Sichuan University, Chengdu 610041, China; liweiqi@stu.scu.edu.cn (W.L.); 2021324030011@stu.scu.edu.cn (Q.S.); 2017151642125@stu.scu.edu.cn (D.Y.); jackpenv2@protonmail.com (J.P.); qmchen@scu.edu.cn (Q.C.)

**Keywords:** periodontitis, micronutrients, dose–response, propensity scores matching, restricted cubic splines

## Abstract

The association between micronutrient intake and the risk of periodontitis has received much attention in recent years. However, most studies focused on the linear relationship between them. This study aimed to explore the dose–response association between micronutrient intake and periodontitis. A total of 8959 participants who underwent a periodontal examination, and reported their micronutrient intake levels were derived from the US National Health and Nutrition Examination Survey (NHANES, 2009–2014) database. Logistic regression was performed to evaluate associations between micronutrient intake and periodontitis after propensity score matching (PSM), and restricted cubic splines (RCS) analysis was conducted to explore the dose–response associations. Following PSM, 5530 participants were included in the RCS analysis. The risk of periodontitis was reduced with sufficient intake of the following micronutrients: vitamin A, vitamin B1, vitamin B2, and vitamin E. In addition, the risk of periodontitis was increased with excessive intake of the following micronutrients: vitamin B1 (1.8 mg/day, males; 1.3 mg/day, females), vitamin C (90 mg/day, males), and copper (1.1 mg/day, combined). In conclusion, a linear association was found between vitamin A, vitamin B2, vitamin C, and copper and periodontitis—namely, a sufficient intake of vitamin A and vitamin B2 might help reduce the prevalence of periodontitis; by contrast, a high intake of vitamin C and copper increased the risk. In addition, a nonlinear dose–response association was found for the incidence of periodontitis with vitamin B1 and vitamin E. When within reasonable limits, supplemental intake helped reduce the prevalence of periodontitis, while excessive intake did not help significantly and might even increase the risk. However, confounding factors, such as health awareness, should still be considered.

## 1. Introduction

Periodontitis is a common chronic inflammatory condition that affects the tissues surrounding the teeth, leading to pocket formation, movement, bone loss, and eventually tooth loss [1,2,3]. In addition to its impact on periodontal tissues, periodontitis is closely associated with the progression of chronic diseases that pose substantial health and financial burden [4,5,6], including diabetes, hypertension, Alzheimer’s disease, and coronary heart disease [7,8,9,10,11]. Prevention and control of the occurrence and development of periodontitis is a pressing concern [12,13,14]. In addition to common risk factors, such as smoking, alcohol consumption, high blood sugar, and high blood pressure [9,15,16,17,18], the close relationship between diet, sleep, mental health, and periodontitis has become a growing research topic in recent years [19,20,21,22,23,24,25,26].

Dietary intake and nutritional supplements are important means for the body to obtain nutrients and maintain normal functioning. The association between micronutrient intake and disease has received widespread attention [27,28,29,30,31,32,33,34,35,36]. Micronutrients [29,30,32,36], which are essential compounds in human metabolism, play extremely important roles in metabolism, growth, development, and health [37,38,39]. Insufficient intake of certain vitamins or minerals can cause physiological disorders and diseases [40,41,42,43]. A previous cross-sectional study conducted in a US population showed that inadequate dietary intake of some micronutrients was statistically significantly associated with the severity of periodontitis [44]. Another study from the same population showed that inadequate dietary calcium intake was strongly associated with periodontitis risk [45]. Differences were likewise found in a cross-sectional study conducted in Denmark, which demonstrated that inadequate dietary calcium intake was associated with periodontitis (in contrast to vitamin D) [46]. A recent review of nutritional studies has highlighted micronutrient deficiencies as risk factors for periodontitis [47,48].

There is substantial evidence supporting a strong association between periodontitis and micronutrient intake [49,50,51,52,53,54,55]. Varied studies have been conducted in different directions to evaluate the association between periodontitis and micronutrient intake [45,56,57,58,59,60]. Existing studies tend to explore the correlation using linear models, showing that associations between micronutrient intake and some other diseases were complex and nonlinear [61,62,63,64,65,66,67]. Despite these findings, we believe that these studies may have overlooked the potential nonlinear dose–response association between micronutrient intake and periodontitis. Therefore, the aim of this study was to screen for micronutrients associated with periodontitis risk using logistic regression after reducing the effect of covariates using propensity score matching (PSM) [68]. Logistic regression and restricted cubic splines (RCS) were used to explore dose–response associations between screened micronutrient intake levels and periodontitis risk.

## 2. Materials and Methods

This study was conducted according to the Strengthening the Reporting of Observational Studies in Epidemiology-Nutritional Epidemiology (STROBE) guidelines [69].

### 2.1. Study Design and Population

The National Health and Nutrition Examination Survey (NHANES) is a cross-sectional study designed to assess the health and nutritional status of adults and children in the United States over time. More specifically, this survey examines a nationally representative sample of approximately 5000 people each year.

The NHANES collects periodontal examination data for those aged >30 years. Herein, we examined subjects with periodontal examination data who participated in three NHANES cycles from 2009–2010 to 2013–2014. Specifically, we included participants who completed a periodontal examination and had two complete records of 24 h dietary intake data during this period. Notably, participants who were pregnant or breastfeeding were excluded [28,29]. Detailed eligibility criteria are shown in the Appendix A. The NHANES dataset was reviewed and approved by the Centers for Disease Control (CDC) and the National Center for Health Statistics (NCHS) ethics review committees, and all subjects provided their written informed consent prior to participation (https://www.cdc.gov/nchs/nhanes/irba98.htm). This study was conducted in accordance with the principles of the Declaration of Helsinki and its later amendments. Details of the relevant data collection and extraction processes can be found at https://www.cdc.gov/nchs/nhanes/index.htm.

### 2.2. Periodontal Examination

A full-mouth periodontal examination was performed on NHANES participants aged ≥30 years between 2009 and 2014. All dental examiners were trained and calibrated by the reference examiner for the survey [70]. Gingival recession and periodontal pocket depths were measured using periodontal probes at six sites on the teeth, and clinical attachment loss was calculated [70]. NHANES researchers referred to the consensus recommendations of the CDC for epidemiological studies and the American Academy of Periodontology Prevention for the diagnosis of periodontitis in conducting this research [71,72].

In the present study, we categorized those with moderate and severe periodontitis into the periodontitis category; those with mild or no periodontitis were grouped into a referent category, to mitigate the risk of bias due to a potentially excessive prevalence of mild periodontitis in the population [73]. Severe/moderate periodontitis was defined as follows:≥2 Interproximal sites with a clinical attachment loss (CAL) of ≥4 mm;≥2 Interproximal sites with a periodontal probing depth of ≥5 mm.

Participants who met one or more of the above two criteria were classified as patients with severe/moderate periodontitis. Referring to previous studies, we excluded edentulous patients and those without complete periodontal examinations [16,74,75].

### 2.3. Micronutrient Intake

The names of the vitamin micronutrients were in a unified format for more convenient statistical analysis and comparison. The vitamins included in the study were vitamin A, vitamin B1 (thiamine), vitamin B2 (riboflavin), vitamin B3 (niacinamide), vitamin B6, vitamin B9 (folic acid), vitamin B12, vitamin C, and vitamin E. The minerals included calcium, copper, iron, magnesium, phosphorus, potassium, selenium, sodium, and zinc.

Micronutrient intake included both dietary intake and dietary supplement intake [30]. Nutritional intake of foods was assessed via 24 h dietary recall at the time of the study interview. Nutritional values were determined using the Food and Micronutrient Database for Dietary Studies (FNDDS, 1.0) [30]. FNDDS provides the nutritional values of each food and beverage reported by NHANES. Calculation of micronutrient intake was based on the type and amount of food consumed.

NHANES participants also reported the dietary supplements they had taken in the past 30 days in an internal interview. For each micronutrient, the daily dose was calculated by combining product information on the frequency of use, ingredients, quantities, and units per serving. The intake of each product was summarized to estimate the total daily dose of each micronutrient for each participant.

The total daily intake of micronutrients was obtained by summing the average of two 24 h dietary data points and the average daily intake of dietary supplements [30]. We also compared our findings to the Dietary Reference Intakes (DRIs), published by the Food and Nutrition Board of the Institute of Medicine in 2006. Inadequate micronutrient intake was defined as a total intake below the recommended dietary allowance (RDA) or adequate intake (AI) levels specified in the DRIs [76]. The classification criteria for inadequate intake are listed in the Appendix A.

### 2.4. Covariates

Study covariates were as follows: age, gender, education level, income, physical activity, smoking status, alcohol intake, obesity, diabetes, hypertension (HTN), and hyperlipidemia (HPL) [74]. Education level was divided into the following categories: those not graduating from high school, high school graduates, those with some college education, and college graduates [75,77]. Income was categorized into three levels based on the household income-to-poverty ratio (IPR): ≤1.30, 1.31–3.50, and >3.50 [34]. Physical activity was classified dichotomously (i.e., as performing physical activity or not performing the physical activity) [34]. Smoking status was divided into never, former, and active smoker categories. Drinkers and nondrinkers were classified according to previously reported criteria [78]. We used body mass index (BMI) as a measure of obesity (in kg/m^2^), according to World Health Organization (WHO) guidelines. Diabetes was defined according to self-reported diabetes (on the study questionnaire) or a glycated hemoglobin level of >6.5%. Patients with systolic blood pressure (SBP) of ≥140 mmHg and/or diastolic blood pressure (DBP) of ≥90 mmHg on physical examination, and/or participants taking antihypertensive medications were considered to have hypertension [79]. Hyperlipidemic patients had a total cholesterol level of ≥240 mg/dL and/or a low-density lipoprotein (LDL) cholesterol level of ≥130 mg/dL. More detailed and additional covariate classifications, along with reference sources, are provided in the Appendix A.

### 2.5. Statistical Analysis

Participants were screened according to the specified study inclusion criteria. We evaluated differences in covariates between participants with moderate/severe periodontitis and those with mild/no periodontitis using rank-sum tests for continuous variables and chi-square tests for categorical variables. Interpolation of missing data using multivariate interpolation of chained equations (MICE, predicting mean matching) was used for continuous variables, and Bayesian multivariate logistic regression was used for categorical variables [80,81].

Logistic regression was used to estimate the associations between vitamin and mineral intake (and whether or not intake was inadequate) and periodontitis. Propensity score matching (PSM) was used to match the characteristics of the study covariates to avoid unbalanced distributions of covariates that could affect the reliability of our findings [68,82].

Four models were built based on adjustments for the different covariates. Model 0 did not adjust for any covariates; Model 1 was a multifactorial logistic regression model adjusting for all 11 covariates mentioned above; Model 2 used PSM to balance covariates, including smoking, alcohol, and physical activity, with multifactorial logistic regression adjusting for age, gender, educational level, income, BMI, diabetes, HPL, and HTN; and Model 3 used PSM to balance all 11 covariates (see Appendix A for details on the adjusted covariates for each model).

Due to the order of magnitude differences in the intake of each micronutrient, the units of intake were converted and adjusted for easy comparison in a logistic regression of continuous variables, as shown in Appendix A. Micronutrients showing statistically significant differences in intake levels (continuous variables) or inadequate intakes (categorical variables) in the PSM-matched single-factor logistic regression were collated, and the stability of the results was tested using multifactorial logistic regression.

More evaluations were conducted on the study data following PSM adjusted for the 11 covariates mentioned above (Model 3). We identified micronutrients with statistically significant differences in intake levels (continuous variables) or inadequate intake (categorical variables) based on PSM-matched single-factor logistic regression. The effect of outliers was excluded using Tukey’s test [83,84]. A multifactorial logistic regression of micronutrient intake was performed and tested for nonlinearity.

In the present study, we used RCS functions to determine the risk points and correlation trends for each micronutrient. An RCS function with adjusted logistic regression was used to analyze dose–response associations between micronutrient intake and periodontitis risk. The results were compared with DRIs, and subgroup analyses were performed according to the reference values provided by the DRIs in reference to different genders or ages.

All data extraction, analysis, and plotting of images were performed using R statistical software (version 4.1.1; The R Project for Statistical Computing, Vienna, Austria). We used the “mice” package to impute the missing values, the “glm” package for logistic regression, the “matching” package for PSM, the “rms” package for RCS, and the “tableone” and “ggplot2” packages for table and graph plotting. Statistical significance was set at an α of <0.05.

## 3. Results

### 3.1. Data Collection

Figure 1 shows the flowchart of participant inclusion. Between 2009 and 2014, a total of 11,753 patients in the NHANES recorded periodontal examination data. After excluding edentulous patients, 10,684 participants had qualifying periodontal examination data. Moreover, 30,468 patients recorded 24 h diet and dietary supplement interview data, of whom 7245 participants recorded fewer than two 24 h dietary interviews. Therefore, dietary intake and dietary supplement data were ultimately included from only 23,223 participants. Moreover, after integrating participants with both periodontal examinations and dietary/dietary supplement intake information and excluding 46 pregnant and 50 breastfeeding participants, 8959 participants were included in the current study in total.

### 3.2. Descriptive Characteristics and Periodontal Disease

Table 1 shows a descriptive summary of the medical and demographic characteristics of the 8959 included participants with periodontitis, compared with those without periodontitis. This study enrolled 4965 participants with no or mild periodontitis and 3994 moderate/severe periodontitis patients. After filling the missing values, missing data were identified in 766 patients with no or mild periodontitis and 607 patients with moderate/severe periodontitis. The statistical results showed higher mean ages and BMI values of patients in the periodontitis group, and that periodontitis was more prevalent in men, as well as in people with lower education and income levels. Smoking, hypertension, hyperlipidemia, and diabetes were strongly associated with the development of periodontitis.

### 3.3. Description of the Four Models Using Logistic Regression and PSM

Statistical analyses were performed separately for each micronutrient intake parameter (continuous variables) and for measures of intake adequacy (categorical variables) according to the four previously designed and described models (Appendix A). The bubble plots in Figure 2A summarize the results of the correlations between micronutrient intake and periodontitis for each model, with the size of the circle representing the magnitude of the *p*-value and the color of the circle representing the odds ratio (OR). Vitamin B1, vitamin B2, vitamin E, and calcium were statistically significantly and negatively correlated with periodontitis risk in all models, and vitamin C was statistically significantly positively correlated with periodontitis risk in Models 2 and 3. The negative associations between vitamin A, vitamin B9, magnesium, phosphorus, and potassium intake and periodontitis became less significant with increasing covariates included in PSM. The results for the categorical variables, reported in Figure 2B, were similar to those for the continuous variables. Associations with vitamin C and calcium intakes were not statistically significant in Model 3, whereas inadequate copper intake was shown to raise the risk of periodontitis.

### 3.4. Characteristics of Micronutrients

Model 3 matched the 11 aforementioned covariates. After matching, 5530 participants were included—half with no or mild periodontitis and the other half with moderate/severe periodontitis. Appendix A presents information on continuous variables representing micronutrient intake after matching the 11 covariates. Appendix A shows the relevant characteristics of the categorical variables. Forest plots were drawn for the continuous and categorical variables after matching (Model 3), and the results are shown in Appendix A. Among the continuous variables, higher intakes of vitamin B1, vitamin B2, vitamin E, and calcium reduced the risk of periodontitis. Higher intake of vitamin C, on the other hand, increased the risk of periodontitis. Among the categorical variables, insufficient intake of vitamin A, vitamin B1, vitamin B2, and copper increased the risk of periodontitis. The results of the multifactorial logistic regression analysis incorporating seven statistically significant micronutrients as described above, including both continuous and categorical variables, were similar to the previously reported one-way logistic regression results (indicating stable model results). The forest plot depicting these data is shown in Figure 3.

### 3.5. Dose–Response Effect Examination

Restricted cubic spline functions incorporated seven micronutrients that differed statistically significantly between categorical or continuous variables. Figure 4 shows the dose–response effect examination results for vitamins A, B1, C, and E, as well as copper. The results for the remaining micronutrients are shown in Appendix A. The overall dose–response effect examination results showed that vitamin A intake was linearly and negatively associated with the risk of periodontitis, and sufficient vitamin A intake reduced the risk of periodontitis (526.7 μg RAE (retinol activity equivalents)/day). Excessive vitamin C intake was linearly and positively correlated with an increased risk of periodontitis; however, vitamin C did not increase the risk of periodontitis if intake was within the recommended intake value (RDA). Vitamin B1, vitamin E, and copper showed complex non-linear relationships. Unlike vitamin E intake, which consistently tended to reduce the risk of periodontitis, a higher intake of vitamin B1 prior to the risk point (1.513 mg/day) reduced the risk of periodontitis. In contrast, an increased intake of vitamin B1 above the risk point increased the risk.

### 3.6. Subgroup Analyses

Figure 5 shows the results of subgroup analyses stratified by age or gender according to the reference threshold of the DRI. For women, adequate vitamin A intake (>491 μg RAE/day) reduced the risk of periodontitis. Associations with vitamin B1 intake were more complex. For males only, an appropriate intake (1.2–1.7 mg/day) was beneficial in reducing the risk of periodontitis, whereas only excess intake (1.3 mg/day) exacerbated the risk for females. Adequate intake of vitamin B2 (2.2 mg/day) in men was beneficial for reducing risk. Additionally, men were found to consume excessive amounts of vitamin C (90 mg/day). More data related to the subgroup analysis can be found in Appendix A.

## 4. Discussion

In the present study, the dose–effect relationship between micronutrients and periodontitis was investigated. We found a nonlinear association between the intake levels of some micronutrients— namely, vitamins B1 and E—and the risk of periodontitis. Our findings suggest that the intake of various micronutrients (such as vitamin B1) within a certain range may be beneficial in reducing the risk of periodontitis, whereas excessive micronutrient intake may increase the risk of periodontitis.

In addition to vitamin B1, multifactorial logistic regression following PSM, and RCS showed that intake within the recommended range of vitamin A and vitamin E levels reduced the risk of periodontitis, while excessive copper intake increased periodontitis risk; this is consistent with the results of previous studies [41,48,51,52]. In addition, our study also showed some results that differed from previous studies; for instance, in our study, excessive vitamin C intake increased periodontitis risk, whereas the results of a previous study conducted in the same population showed vitamin C to be a promoter of periodontal health [41,48].

We note that, in the current study, many micronutrient intake deficiencies were associated with a higher incidence of periodontitis when not adjusted for covariates. However, with progressively increasing adjustments for the 11 covariates described above, these statistically significant associations largely became statistically insignificant. We suggest that the statistically significant correlations present in the unadjusted data may arise from the influence of covariates, thus exaggerating associations between micronutrients and periodontitis; this could potentially explain the contradictory results between our study and those reported in previous investigations [44,45,46,47].

Notably, copper intake reduced the risk of periodontitis with regard to the evaluated categorical variables and likewise increased the risk of periodontitis in multivariate logistic regression analyses incorporating continuous variables. Both findings were consistent with our RCS results. This shows the limitations of using purely categorical variables, which may yield different or even opposite results if dose–response correlations are not explored. It is noteworthy that insufficient intake of both vitamin A and vitamin E occurs in over 50% of the population, and the detected associations between supplementation of these two vitamins and the prevention and treatment of periodontitis need to be taken very seriously.

In our subgroup analyses, threshold evaluations and nonlinear relationships differed according to gender and age, and we believe that the effect of the intake of different nutrients on the risk of periodontitis differs between genders or age groups (based on our work and the results of prior studies) [51,83]. Additional studies are needed to investigate this question more comprehensively. In addition, there was a trend of direct correlations between the intake levels of some micronutrients and periodontitis (e.g., for vitamin B2 intake in women), but the ORs and associated confidence intervals were not considered statistically significant. This may be due to our modest sample size, and therefore, a study enrolling a larger population is needed to determine this correlation trend.

Given the results of this and previous studies, we suggest that micronutrients are closely related to the biological mechanisms of periodontitis through mechanistic pathways, including those related to oxidative stress, inflammation, collagen structure, and bone mineralization. Since many micronutrients have roles in innate and adaptive immune responses (in particular, playing crucial roles in the immune response to oral biofilms), there is great potential with regard to nutritional research in the field of periodontitis prevention and treatment [48,67]. The potential risks and benefits of micronutrients in relation to periodontal health need to be evaluated more comprehensively in future research.

Our study has the following strengths. First, it is the first such study conducted to date that used RCS to explore the correlation between periodontitis and micronutrient intake. This breaks the tradition of evaluating simplistic “favorable” or “harmful” relationships between periodontitis and vitamin intake. Instead, we extended prior research by investigating favorable ranges of micronutrient intake levels as well as evaluating the threshold above which intake is harmful. This has helped in exploring these complex associations more deeply. Second, our study data were based on a nationally representative sample of US adults and were collected using a validated study design and methodology, including adjusting for 11 covariates to exclude as many confounding factors as possible from our evaluation. Third, the present study, which builds on previous studies, is the first investigation to explore the association between micronutrients and periodontitis using a randomization design performed with PSM to better correct for confounding factors (such as selection bias). Finally, we used both continuous variables (representing total micronutrient intake) and categorical variables (representing intake adequacy) to screen for eligible (i.e., statistically significant) micronutrients before using RCS to explore trends in correlations to make the interpretation of our results more reliable.

In addition to these strengths, this study has several limitations. First, dietary supplement intake was averaged to a daily amount using the results of a recall interview from the previous 30 days and daily food intake based on NHANES, which incorporates two 24 h dietary recalls. This may not reflect the actual daily intake. In addition, micronutrient intake was self-reported; therefore, micronutrient levels in blood samples are needed for future analysis. Second, although the use of PSM reduced the bias of estimates, a larger sample size is needed. Third, a cross-sectional design is inadequate for establishing causality. As such, a cohort study is needed in the future. Fourth, our study did not consider confounding factors of health awareness on outcomes. This is important as participants’ different health awareness may have influenced their micronutrient intake. Finally, after grouping and implementing RCS according to gender and age, we found that micronutrients showing an overall linear relationship may show a nonlinear relationship with periodontitis after grouping and that the opposite trend occurs as well; therefore, our findings should be interpreted with caution.

## 5. Conclusions

This study found a significant association between vitamin B1, vitamin B2, vitamin C, vitamin A, vitamin E, and copper and periodontitis. To the best of our knowledge, this was the first study to report a nonlinear dose–response for the incidence of periodontitis with vitamin B1 and vitamin E. When within reasonable limits, supplemental intake helped reduce the prevalence of periodontitis; however, excessive intake did not significantly help and might even increase the risk. In addition, there was a linear association between vitamin A, vitamin B2, vitamin C, and copper and periodontitis, where sufficient intake of vitamin A and vitamin B2 helped reduce the prevalence of periodontitis. However, high intakes of vitamin C and copper increased the risk. Adjusting micronutrient intake might help prevent and treat periodontitis.

## Figures and Tables

**Figure 1 nutrients-14-02466-f001:**
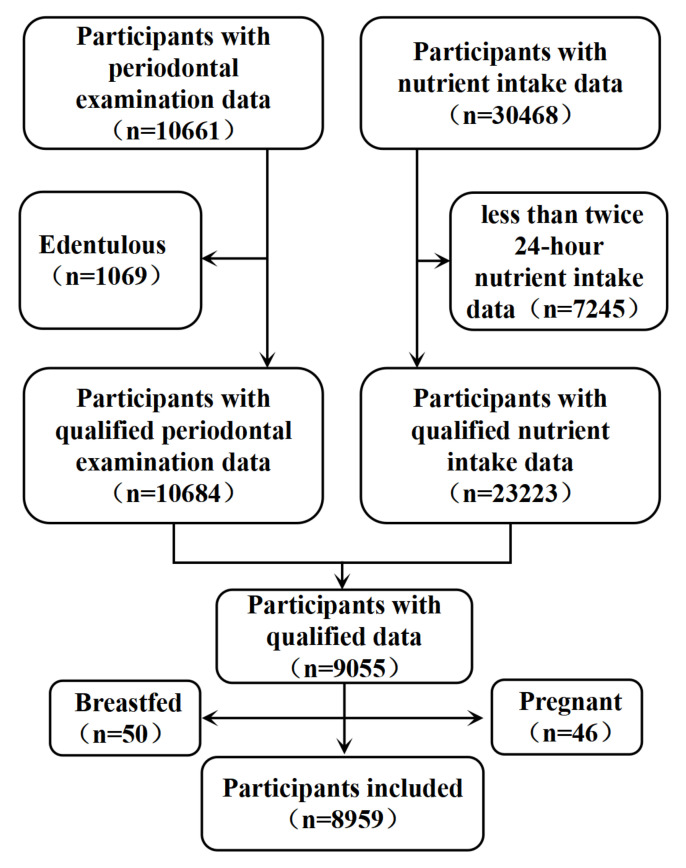
Flowchart for screening participants.

**Figure 2 nutrients-14-02466-f002:**
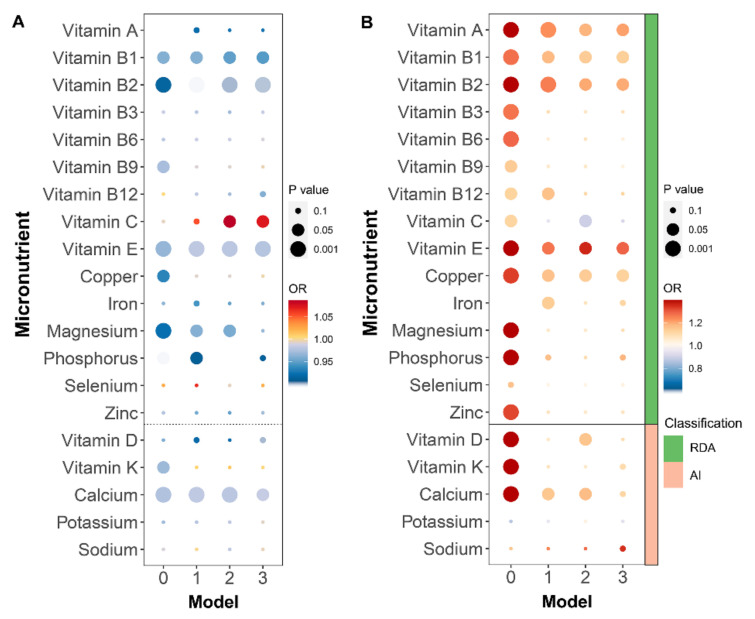
Logistic regression results for categorical and continuous variables for the four models: (**A**) logistic regression results of micronutrient intake (continuous variable); (**B**) logistic regression results of intake inadequate (categorical variable). RDA: recommended dietary allowance; AI: adequate intake. Model 0 had no adjustment of data; Model 1 was a multifactor logistic regression adjusted for all eleven covariates; Model 2 used propensity score matching to be balanced covariates (including smoke, alcohol, and physical activity) and then used multifactorial logistic regression adjusting for age, gender, educational level, income, BMI, diabetes, HPL, and HTN; Model 3 used PSM to balance all eleven covariates.

**Figure 3 nutrients-14-02466-f003:**
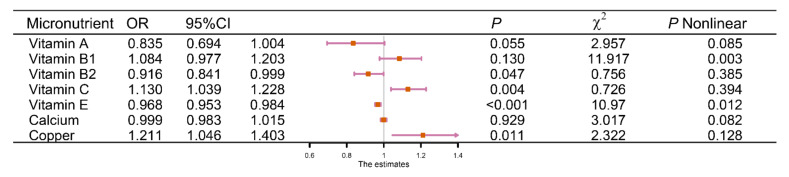
Result of the multifactorial logistic regression of 8 significantly micronutrients.

**Figure 4 nutrients-14-02466-f004:**
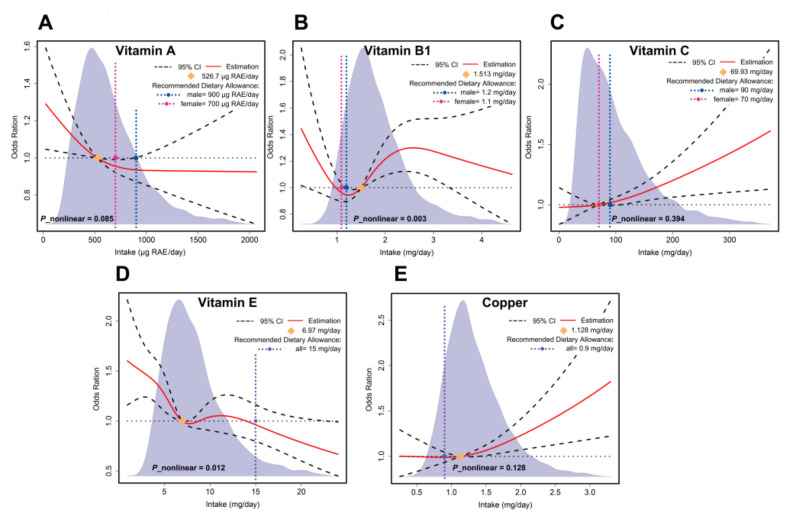
The RCS results for the micronutrients: (**A**) RCS results for vitamin A; (**B**) vitamin B1; (**C**) vitamin C; (**D**) vitamin E; (**E**) copper.

**Figure 5 nutrients-14-02466-f005:**
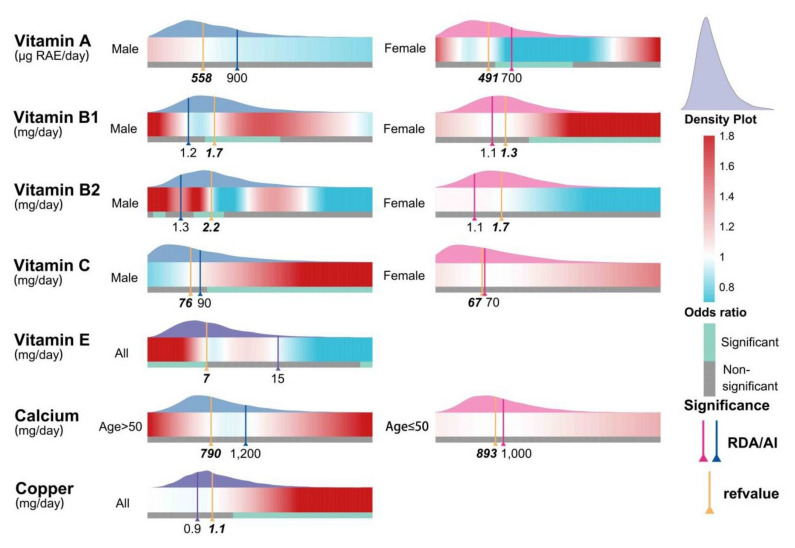
The RCS results after subgroup analysis for age or gender according to the reference threshold of DRI. RAE: retinol activity equivalents. Significance: intervals for the intake of significant micronutrients were determined based on odds ratio and confidence intervals. RDA: recommended dietary allowance; AI: adequate intake. Refvalue: the risk point.

**Table 1 nutrients-14-02466-t001:** Characteristic of covariant.

Level		Overall ^a^	Periodontitis	No Periodontitis ^a^	χ^2^	*p* ^b^
No.		8959	3994	4965		
Age (mean (SD))		52.36 (14.16)	56.75 (13.63)	48.82 (13.59)	/	<0.001
Gender (%)	Female	4585 (51.2)	1681 (42.1)	2904 (58.5)	237.64	<0.001
	Male	4374 (48.8)	2313 (57.9)	2061 (41.5)		
Edu (%)	<High school graduate	1950 (21.8)	1209 (30.3)	741 (14.9)	551.05	<0.001
	College	2576 (28.8)	1053 (26.4)	1523 (30.7)		
	College graduate	2490 (27.8)	717 (18.0)	1773 (35.7)		
	High school graduate	1943 (21.7)	1015 (25.4)	928 (18.7)		
Income (%)	High	3106 (34.7)	1002 (25.1)	2104 (42.4)	314.75	<0.001
	Low	2600 (29.0)	1419 (35.5)	1181 (23.8)		
	Middle	3253 (36.3)	1573 (39.4)	1680 (33.8)		
BMI (mean (SD))		29.48 (6.74)	29.65 (6.73)	29.35 (6.74)	/	0.037
Diabetes (%)	Diabetes	1399 (15.6)	845 (21.2)	554 (11.2)	167.17	<0.001
	No diabetes	7560 (84.4)	3149 (78.8)	4411 (88.8)		
Alcohol (%)	Drinkers	6604 (73.7)	2936 (73.5)	3668 (73.9)	0.1354	0.713
	Nondrinkers	2355 (26.3)	1058 (26.5)	1297 (26.1)		
HPL (%)	Hyperlipidemia	4077 (45.5)	1939 (48.5)	2138 (43.1)	26.646	<0.001
	No hyperlipidemia	4882 (54.5)	2055 (51.5)	2827 (56.9)		
HTN (%)	Hypertension	5113 (57.1)	2612 (65.4)	2501 (50.4)	203.35	<0.001
	No hypertension	3846 (42.9)	1382 (34.6)	2464 (49.6)		
Phy (%)	No physical activity	2866 (32.0)	1406 (35.2)	1460 (29.4)	33.923	<0.001
	Physical activity	6093 (68.0)	2588 (64.8)	3505 (70.6)		
Smoke (%)	Active smoker	1598 (17.8)	970 (24.3)	628 (12.6)	334.72	<0.001
	Former smoker	2322 (25.9)	1180 (29.5)	1142 (23.0)		
	Never smoker	5039 (56.2)	1844 (46.2)	3195 (64.4)		

^a^: Continuous variables such as age and gender are the mean (standard deviation), and other categorical variables are the number of people (percentage); ^b^: chi-square test was used for continuous variables and rank-sum test for classified variables; SD: stand error; BMI: body mass index; HPL: hyperlipidemia; HTN: hypertension; phy: physical activity.

## Data Availability

The data used for this study, though not available in a public repository, will be made available to other researchers upon reasonable request.

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
