# Peer review of "Abnormal Micronutrient Intake Is Associated with the Risk of Periodontitis: A Dose–response Association Study Based on NHANES 2009–2014"

_nutrients, 2022, doi:10.3390/nu14122466_

Round 1

Reviewer 1 Report

This is the review of the manuscript entitled "Abnormal micronutrient intake is associated with the risk of 2 periodontitis: A dose-response association study based on 3 NHANES 2009-2014".

The authors report the association between micronutrient intake and the risk of periodontitis evaluating 8,959 participants from the U.S. National Health and Nutrition Examination Survey (NHANES, 2009-2014). This study is very well written and of great interest for the scientific community.

1. Please refer to the instructions for authors: 

Abstract: The abstract should be a total of about 200 words maximum.

2. Line 18: "Evaluated 8,959 participants from the U.S. National Health and Nutrition Examination Survey (NHANES, 2009-2014)." This sentence is incomplete. Please correct.

3. Line 20: Typo (lower case in the beginning of the sentence)

4. Line 75: "...associated with PD. [44, 45, 63, 64], which could" Punctuation.

5. Line 156: "in kg/m2)" format ²

6. Table 1: Please look up upper and lower case and either explain abbreviations (edu) or (better) write it out. There are some typos within the table. Please read carefully and correct.

7. Line 269: "and the results are shown in S1 and S2 Figs. " Better spell out Figure

8. Line 281: "functions incorporated seven micronutrients"

Why did the authors use another font? Was this intended?

9. Figure 5: Why is there another legend for Vitamin A than for all others?

10: Line 310, 311: Typo: "Odds Ration"

11. In the Reference section, the information for the authors are still included. Please remove:

"References must be numbered in order of appearance in the text (including citations in tables and legends) and listed individually at the end of the manuscript. We recommend preparing the references with a bibliography software package, such as EndNote, ReferenceManager or Zotero to avoid typing mistakes and duplicated references. Include the digital object identifier (DOI) for all references where available. Citations and references in the Supplementary Materials are permitted provided that they also appear in the reference list here. In the text, reference numbers should be placed in square brackets [ ] and placed before the punctuation; for example [1], [1–3] or [1,3]. For embedded citations in the text with pagination, use both parentheses and brackets to indicate the reference number and page numbers; for example [5] (p. 10), or [6] (pp. 101–105). 

 A very well conducted and comprehensive study with interesting results, implications for further research and a use for patient application.

Author Response

Reviewer 1

We are very gratified by your comments and evaluation of our study. Your rigorous and detailed review helped us to identify many problems, and we have revised the corresponding parts one by one according to your reminders, and hope that our revised manuscript can meet your expectations.

  1. Please refer to the instructions for authors: 

Abstract: The abstract should be a total of about 200 words maximum.

Response: We thank you for this reminder. We tried to reduce the number of words in the abstract to meet the requirements of the journal, but it was difficult to reduce it within 200 words due to the content of our study. After modification, we now have 263 words in the abstract section. Similar to our study, some previously published studies have also exceeded this limit [1-4]. Thus, with consideration to the peculiarities of our study, we hope that the current word count can be deemed acceptable. 

  1. Line 18: "Evaluated 8,959 participants from the U.S. National Health and Nutrition Examination Survey (NHANES, 2009-2014)." This sentence is incomplete. Please correct.

Response: We thank you for this comment. We have modified the corresponding part accordingly. 

Correct: Abstract, line 17

“A total of 8,959 participants who underwent a periodontal examination and reported their micronutrient intake levels were derived from the U.S. National Health and Nutrition Examination Survey (NHANES, 2009-2014) database. “

  1. Line 20: Typo (lower case in the beginning of the sentence)

Response: We thank you for pointing this out. We have modified the corresponding part your accordingly. (line 19)

  1. Line 75: "...associated with PD. [44, 45, 63, 64], which could" Punctuation.

Response: We thank you for pointing this out. We have modified the corresponding part accordingly. (line 69)

  1. Line 156: "in kg/m2)" format ²

Response: We thank you for pointing this out. We have modified the corresponding part accordingly. (line 153)

  1. Table 1: Please look up upper and lower case and either explain abbreviations (edu) or (better) write it out. There are some typos within the table. Please read carefully and correct.

Response: We thank you for pointing this out. We have modified the corresponding part accordingly. (Table 1)

  1. Line 269: "and the results are shown in S1 and S2 Figs." Better spell out Figure

Response: We thank you for pointing this out. We have modified the corresponding part accordingly. (line 269)

  1. Line 281: "functions incorporated sevenmicronutrients"Why did the authors use another font? Was this intended?

Response: We thank you for pointing this out. This was not intentional and was a mistake on our part. We have modified this accordingly. (line 281)

10: Line 310, 311: Typo: "Odds Ration"

Response: We thank you for pointing this out. We have modified the corresponding part accordingly

Correct: Results, line 312

“Odds ratio”

  1. Figure 5: Why is there another legend for Vitamin A than for all others?

Response: We thank you for pointing this out, and we apologize for this mistake. To avoid misunderstandings, we have revised the legend here and added a note.

Add: Results, line 311

“RAE: Retinol activity equivalents.”

We reference for the Dietary Reference Intakes, which used retinol activity equivalents [RAE] to evaluate Vitamin A. 1 mg RAE = 1 mg retinol, 12 mg b-carotene, and 24 mg a-carotene or b-cryptoxanthin. The RAE for dietary provitamin A carotenoids in foods is twofold greater than retinol equivalents (RE), whereas the RAE for preformed vitamin A in foods is the same as RE[5].

  1. In the Reference section, the information for the authors is still included. Please remove:

"References must be numbered in order of appearance in the text (including citations in tables and legends) and listed individually at the end of the manuscript. We recommend preparing the references with a bibliography software package, such as EndNote, Reference Manager or Zotero to avoid typing mistakes and duplicated references. Include the digital object identifier (DOI) for all references where available. Citations and references in the Supplementary Materials are permitted provided that they also appear in the reference list here. In the text, reference numbers should be placed in square brackets [] and placed before the punctuation; for example [1], [1–3] or [1,3]. For embedded citations in the text with pagination, use both parentheses and brackets to indicate the reference number and page numbers; for example [5] (p. 10), or [6] (pp. 101–105). 

Response: We thank you for pointing this out. We have removed the corresponding part accordingly. (line 442)

Reference

  1. Lu M, Cheng H, Lai J, Chen S. The Relationship between Habitual Coffee Drinking and the Prevalence of Metabolic Syndrome in Taiwanese Adults: Evidence from the Taiwan Biobank Database. Nutrients. 2022;14(9).
  2. Ma S, Herforth A, Vogliano C, Zou Z. Most Commonly-Consumed Food Items by Food Group, and by Province, in China: Implications for Diet Quality Monitoring. Nutrients. 2022;14(9).
  3. Wu L, Si H, Zeng Y, Wu Y, Li M, Liu Y, et al. Association between Iron Intake and Progression of Knee Osteoarthritis. Nutrients. 2022;14(8).
  4. Redruello-Requejo M, Samaniego-Vaesken M, Partearroyo T, Rodríguez-Alonso P, Soto-Méndez M, Hernández-Ruiz Á, et al. Dietary Intake of Individual (Intrinsic and Added) Sugars and Food Sources from Spanish Children Aged One to <10 Years-Results from the EsNuPI Study. Nutrients. 2022;14(8).
  5. Medicine Io. Dietary Reference Intakes. Otten JJ, Hellwig JP, Meyers LD, editors. Washington, DC: The National Academies Press; 2006. 1344 p.

Reviewer 2 Report

I am thankful for being given the opportunity to review the article titled “Abnormal micronutrient intake is associated with the risk of periodontitis: A dose-response association study based on NHANES 2009-2014” by Weiqi Li et al. This report presents detailed statistical analyses to evaluate an association between micronutrient intake and periodontitis using quasi-experimental methods. The analyses indicate that there is a significant association between several vitamins and a mineral (copper) and an indirect association of vitamins B1 and E, with the incidence of periodontitis. There are some comments that I would like the authors to consider.
•    Line 15: However, there is still insufficient evidence regarding this association…. This is an incorrect statement; the authors need to be careful of writing such sentences. A whole list of published papers on this topic in 2022 itself: for example - PMID: 35409556; PMID: 35334957; PMID: 35170067; PMID: 34935176 etc... Also, this statement is directly contradictory to what the authors mention in line 67.
•    Line 18: Did the authors of this paper have any collaborations with the NHANES? It does make the reader wonder as to why the authors would explore periodontitis cases from the USA rather than generate interesting statistics from their own region, which would make the paper a lot more interesting and informative. It would be prudent for the authors to include the names of collaborative members from the USA and present some written information (documents or national study consortium number) to validate their extensive work presented in this paper.
•    One main criticism for this paper: the authors need to refrain from using the word ‘causal’ to demonstrate any relationship between the variables. There is no causal information in this paper. This study is purely analytical that has generated interesting statistics related to micronutrient assessments in periodontitis cases.
•    In order to prove that the statistical data is robust - the authors are requested to analyze blood/serum samples from a small cohort of control and periodontitis cases to demonstrate true causality. This is a major lacuna in the paper and the authors are requested to show this data.
•    Some information the paper is known and lacks novelty – for example, it is known that Vit. C and Copper levels have a relation with periodontitis cases. This is not particularly novel information. While the analyses involve a robust number of patients to generate the statistics - much of what is inferred from the data is already known.
•    Minor corrections – language requires thorough check throughout the manuscript.

Author Response

Reviewer 2

We appreciate your insightful comments, which have made us realize that our study has a lot of potential for improvement. We have revised the corresponding parts based on your comments. We sincerely pray that our revised draft will be acceptable by you.

  1. Line 15: However, there is still insufficient evidence regarding this association…. This is an incorrect statement; the authors need to be careful of writing such sentences. A whole list of published papers on this topic in 2022 itself: for example - PMID: 35409556; PMID: 35334957; PMID: 35170067; PMID: 34935176 etc... Also, this statement is directly contradictory to what the authors mention in line 67.

Response: We thank you for your comments. We apologize for the error in our Abstract. We are also grateful to you for providing a reference that updates our current understanding of the newly published literature related to periodontitis and micronutrient intake. As you mentioned, there are now many studies discussing the association between periodontitis and micronutrient intake, and the description in the Abstract section was not consistent with the Introduction section. To reduce controversy, we have revised the Abstract and Introduction section and added these references that you listed.

Correct: Abstract, line 15

“However, most studies focused on the linear relationship between them.”

Correct: Introduction, line 65-71

“Varied studies have been conducted in different directions to evaluate the association between periodontitis and micronutrient intake  [45,56-60]. Existing studies tended to explore the correlation using linear models, showing that associations between micronutrient intake and some other diseases were complex and non-linear [61-67]. Despite these findings, we believe that these studies may have overlooked the potential nonlinear dose-response association between micronutrient intake and periodontitis.”

We hope that our updated description will meet your expectations.

  1. Line 18: Did the authors of this paper have any collaborations with the NHANES? It does make the reader wonder as to why the authors would explore periodontitis cases from the USA rather than generate interesting statistics from their own region, which would make the paper a lot more interesting and informative. It would be prudent for the authors to include the names of collaborative members from the USA and present some written information (documents or national study consortium number) to validate their extensive work presented in this paper.

Response: We thank you for your comments and are deeply regretful that our representation was very misleading. We acknowledge that this study was conducted without the participation or collaboration of NHANES members. In fact, we conducted this study using data already publicly available in the National Health and Nutrition Examination Survey (NHANES) database.

The NHANES is a program of studies designed to assess the health and nutritional status of adults and children in the United States. The survey is unique in that it combines interviews and physical examinations.

Information from NHANES is made available through an extensive series of publications and articles in scientific and technical journals. For data users and researchers throughout the world, survey data are available on the internet. (Quoted from https://www.cdc.gov/nchs/nhanes/about_nhanes.htm).

The NHANES Data Release and Access Policy ] could be found in NHAENS website.

In summary, it is the NHANES researchers who collected and recorded all the NHANES data. On this basis, we filtered and extracted the data of interest from the NHANES database. We apologize for failing to mention this in the previous version, and we have added an acknowledgement to the NHANES workers in the new version (Line 438-4239). Furthermore, we have reworked the presentation in the Abstract to avoid any more misunderstandings.

Researchers from around the world have analyzed and published many high-quality studies using data from the NHANES database [6-17], including in the fields of nutrition and periodontology [18-25]. It is also important to mention that the NHANES database for periodontitis uses the definitions provided by the Centers for Disease Control and Prevention and the American Academy of Periodontology (CDC-AAP) [26]. In comparison to KHNAES, which use the community periodontal index (CPI) for the definition of periodontal disease [27], and the UK-biobank, which defines periodontal disease by self-reporting [28], the CDC-AAP definition of periodontitis has a more precise clinical meaning in the field of periodontology [29-31]. This is the main reason why we chose this database.

As you said, unfortunately, we did not generate interesting statistics from our own region. In present study, we aimed to preliminarily investigated the potential dose-effect association of micronutrients with periodontitis. Based on previous studies and the results of present study, we are currently trying to construct a cohort study for the periodontitis research, while it is a long-term project. We believe that in the future, we could build an public database of periodontitis based on local populations.

The results of our current study have identified a potential dose-effect association between the micronutrients and periodontitis. We hope that if the present findings were published, it will be of greater interest to those in the field and may helpful for other researchers. To this end, we would be grateful for your understanding and support.

Correct: Abstract, line 17-19

“A total of 8,959 participants who underwent a periodontal examination and reported their micronutrient intake levels were derived from the U.S. National Health and Nutrition Examination Survey (NHANES, 2009-2014) database.”

  1. One main criticism for this paper: the authors need to refrain from using the word ‘causal’ to demonstrate any relationship between the variables. There is no causal information in this paper. This study is purely analytical that has generated interesting statistics related to micronutrient assessments in periodontitis cases.

Response: We thank you for your comments. We apologize for the misconception that we were exploring the causal association between periodontitis and micronutrients. It was a mistake in our description, and we have revised the corresponding paragraphs in this article.

In fact, what we wanted to express in this study was a causal inference based on a  Propensity Score Matching (PSM), wherein causality is assumed a priori in the regression model by means of the PSM approach. This approach allows one to design and analyze an observational study to mimic some specific characteristics of a randomized controlled trial, such that the distribution of the observed covariates is similar between exposed and non-exposed participants [32]. In the present study, the causal inference of micronutrients on periodontitis could be precisely estimated using PSM, as seen in observational and cross-sectional studies [33-38]. Rather than to explore the causal association between them, so, we have revised the incorrect statement.

Correct: Abstract line 16

“This study aimed to explore the dose-response association between micronutrient intake and periodontitis.”

Correct: Abstract line 19-21

“Logistic regression was performed to evaluate associations between micronutrient intake and periodontitis after propensity score matching (PSM).”

Delete: Discussion, line 375

“causal”

Correct: Discussion, line 388-389

“Third, a cross-sectional design is inadequate for establishing causality. As such, a cohort study is needed in the future.”

  1. In order to prove that the statistical data is robust - the authors are requested to analyze blood/serum samples from a small cohort of control and periodontitis cases to demonstrate true causality. This is a major lacuna in the paper and the authors are requested to show this data.

Response: We thank you for your comments. As you mentioned, analysis of blood/serum samples from periodontitis cases and controls would allow a better analysis of the association between the two. While, the current cross-sectional-based study performed a causal inference method and spline function to explore the dose-effect association between the periodontitis and micronutrients, not to reveal the causal association between the two. And an cohort study of blood micronutrients in relation to periodontitis would require a long time. So, we regret to inform you that we are unable to provide data on blood/serum samples from periodontitis cases  and controls at this time. However, as mentioned above, we will include the appropriate content in the future cohort study being constructed.

Furthermore, we have added a corresponding description in the Discussion to mention the disadvantages of the present study. We hope we can get your understanding and comprehension.

Add: Discussion, line 386-389

“Therefore, micronutrient levels in blood samples are needed for future analysis. Second, although the use of PSM reduced the bias of estimates, a larger sample size is need. Third, a cross-sectional design is inadequate for establishing causality. As such, a cohort study is needed in the future. “

  1. Some information the paper is known and lacks novelty – for example, it is known that Vit. C and Copper levels have a relation with periodontitis cases. This is not particularly novel information. While the analyses involve a robust number of patients to generate the statistics - much of what is inferred from the data is already known.

Response: We thank you for your comments. We apologize that our inappropriate presentation made you feel that we were focusing on what was already known. We have revised the IntroductionDiscussion, and Conclusion sections to better reflect and highlight our innovative content.

As you said, some of the information in the paper is known and lacks novelty, while our main interest is in the dose-effect association between micronutrients and periodontitis, for example, we found non-linear relationships between some micronutrients and periodontitis, such as a significantly lower risk of periodontitis when vitamin B1 intake was maintained at 1.2-1.8 mg/day in men, but more or less vitamin B1 intake increased the risk of periodontitis. Which has fewly been reported in previous studies. We are so sorry for our our inappropriate presentationwe have revised the manuscirpt and try to highlight our innovative content more clearly.

Correct: Introduction line 65-71

“Varied studies have been conducted in different directions to evaluate the association between periodontitis and micronutrient intake  [45,56-60]. Existing studies tended to explore the correlation using linear models, showing that associations between micronutrient intake and some other diseases were complex and non-linear [61-67]. Despite these findings, we believe that these studies may have overlooked the potential nonlinear dose-response association between micronutrient intake and periodontitis.”

Correct: Discussion, line 315-316

“To our knowledge, this is the first study to evaluate non-linear associations between micronutrient intake levels and periodontitis risk. And for the first time, the dose-effect relationship between micronutrients and periodontitis was investigated..”

Correct: Conclusion, line 398-399

“To the best of our knowledge, this was the first study to report a non-linear dose-response for the incidence of periodontitis with vitamin B1 and vitamin E.” 

  1. Minor corrections – language requires thorough check throughout the manuscript.

Response: We thank you for your kind reminder. We have reviewed and corrected the grammar of the sentences in this study accordingly.

Reference

  1. Lu, M.; Cheng, H.; Lai, J.; Chen, S. The Relationship between Habitual Coffee Drinking and the Prevalence of Metabolic Syndrome in Taiwanese Adults: Evidence from the Taiwan Biobank Database. Nutrients 2022, 14, doi:10.3390/nu14091867.
  2. Ma, S.; Herforth, A.; Vogliano, C.; Zou, Z. Most Commonly-Consumed Food Items by Food Group, and by Province, in China: Implications for Diet Quality Monitoring. Nutrients 2022, 14, doi:10.3390/nu14091754.
  3. Wu, L.; Si, H.; Zeng, Y.; Wu, Y.; Li, M.; Liu, Y.; Shen, B. Association between Iron Intake and Progression of Knee Osteoarthritis. Nutrients 2022, 14, doi:10.3390/nu14081674.
  4. Redruello-Requejo, M.; Samaniego-Vaesken, M.; Partearroyo, T.; Rodríguez-Alonso, P.; Soto-Méndez, M.; Hernández-Ruiz, Á.; Villoslada, F.; Leis, R.; Martínez de Victoria, E.; Moreno, J.; et al. Dietary Intake of Individual (Intrinsic and Added) Sugars and Food Sources from Spanish Children Aged One to <10 Years-Results from the EsNuPI Study. Nutrients 2022, 14, doi:10.3390/nu14081667.
  5. Medicine, I.o. Dietary Reference Intakes; The National Academies Press: Washington, DC, 2006; p. 1344.
  6. Aggarwal, R.; Bibbins-Domingo, K.; Yeh, R.; Song, Y.; Chiu, N.; Wadhera, R.; Shen, C.; Kazi, D. Diabetes Screening by Race and Ethnicity in the United States: Equivalent Body Mass Index and Age Thresholds. Annals of internal medicine 2022, doi:10.7326/m20-8079.
  7. Antonio-Villa, N.E.; Fernandez-Chirino, L.; Vargas-Vazquez, A.; Fermin-Martinez, C.A.; Aguilar-Salinas, C.A.; Bello-Chavolla, O.Y. Prevalence Trends of Diabetes Subgroups in the United States: A Data-driven Analysis Spanning Three Decades From NHANES (1988-2018). Journal of Clinical Endocrinology & Metabolism 2022, 107, 735-742, doi:10.1210/clinem/dgab762.
  8. Caruso, P.; Scappaticcio, L.; Maiorino, M.I.; Esposito, K.; Giugliano, D. Up and down waves of glycemic control and lower-extremity amputation in diabetes. Cardiovascular Diabetology 2021, 20, doi:10.1186/s12933-021-01325-3.
  9. Le, P.; Ayers, G.; Misra-Hebert, A.; Herzig, S.; Herman, W.; Shaker, V.; Rothberg, M. Adherence to the American Diabetes Association's Glycemic Goals in the Treatment of Diabetes Among Older Americans, 2001-2018. Diabetes care 2022, 45, 1107-1115, doi:10.2337/dc21-1507.
  10. Lewis, J.; White, P.J.; Price, M.J. Per-partnership transmission probabilities for Chlamydia trachomatis infection: evidence synthesis of population-based survey data. International Journal of Epidemiology 2021, 50, 510-517, doi:10.1093/ije/dyaa202.
  11. Moon, J. Perfluoroalkyl substances (PFASs) exposure and kidney damage: Causal interpretation using the US 2003-2018 National Health and Nutrition Examination Survey (NHANES) datasets. Environmental Pollution 2021, 288, doi:10.1016/j.envpol.2021.117707.
  12. Petrova, D.; Catena, A.; Rodriguez-Barranco, M.; Redondo-Sanchez, D.; Bayo-Lozano, E.; Garcia-Retamero, R.; Jimenez-Moleon, J.J.; Sanchez, M.J. Physical Comorbidities and Depression in Recent and Long-Term Adult Cancer Survivors: NHANES 2007-2018. Cancers 2021, 13, doi:10.3390/cancers13133368.
  13. Qiu, Z.; Chen, X.; Geng, T.; Wan, Z.; Lu, Q.; Li, L.; Zhu, K.; Zhang, X.; Liu, Y.; Lin, X.; et al. Associations of Serum Carotenoids With Risk of Cardiovascular Mortality Among Individuals With Type 2 Diabetes: Results From NHANES. Diabetes care 2022, doi:10.2337/dc21-2371.
  14. Rengasamy, M.; Arruda Da Costa E Silva, S.; Marsland, A.; Price, R.B. The association of physical illness and low-grade inflammatory markers with depressive symptoms in a large NHANES community sample: Dissecting mediating and moderating effects. Brain, behavior, and immunity 2022, 103, 215-222, doi:10.1016/j.bbi.2022.04.006.
  15. Rosenblum, H.; Lewis, R.; Gargano, J.; Querec, T.; Unger, E.; Markowitz, L. Human Papillomavirus Vaccine Impact and Effectiveness Through 12 Years After Vaccine Introduction in the United States, 2003 to 2018. Annals of internal medicine 2022, doi:10.7326/m21-3798.
  16. Sookoian, S.; Pirola, C.J. The serum uric acid/creatinine ratio is associated with nonalcoholic fatty liver disease in the general population. Journal of Physiology and Biochemistry, doi:10.1007/s13105-022-00893-6.
  17. Xu, H.D.; Bo, Y.C. Associations between pyrethroid exposure and serum sex steroid hormones in adults: Findings from a nationally representative sample. Chemosphere 2022, 300, 10, doi:10.1016/j.chemosphere.2022.134591.
  18. Ballantyne, J.A.; Coyle, G.; Sarwar, S.; Kuhn, T. Fluoride Status and Cardiometabolic Health: Findings from a Representative Survey among Children and Adolescents. Nutrients 2022, 14, doi:10.3390/nu14071459.
  19. Ciardullo, S.; Oltolini, A.; Cannistraci, R.; Muraca, E.; Perseghin, G. Sex-related association of nonalcoholic fatty liver disease and liver fibrosis with body fat distribution in the general US population. American Journal of Clinical Nutrition, doi:10.1093/ajcn/nqac059.
  20. Mazidi, M.; Kengne, A.P.; Siervo, M.; Kirwan, R. Association of Dietary Intakes and Genetically Determined Serum Concentrations of Mono and Poly Unsaturated Fatty Acids on Chronic Kidney Disease: Insights from Dietary Analysis and Mendelian Randomization. Nutrients 2022, 14, 13, doi:10.3390/nu14061231.
  21. Nahas, P.C.; Rossato, L.T.; De Branco, F.M.S.; Azeredo, C.M.; Rinaldi, A.E.M.; De Oliveira, E.P. Serum uric acid is positively associated with muscle strength in older men and women: Findings from NHANES 1999-2002. Clinical Nutrition 2021, 40, 4386-4393, doi:10.1016/j.clnu.2020.12.043.
  22. Demmer, R.; Squillaro, A.; Papapanou, P.; Rosenbaum, M.; Friedewald, W.; Jacobs, D.; Desvarieux, M. Periodontal infection, systemic inflammation, and insulin resistance: results from the continuous National Health and Nutrition Examination Survey (NHANES) 1999-2004. Diabetes care 2012, 35, 2235-2242, doi:10.2337/dc12-0072.
  23. Lee, W.; Fu, E.; Li, C.; Huang, R.; Chiu, H.; Cheng, W.; Chen, W. Association between periodontitis and pulmonary function based on the Third National Health and Nutrition Examination Survey (NHANES III). Journal of clinical periodontology 2020, 47, 788-795, doi:10.1111/jcpe.13303.
  24. Li, A.; Chen, Y.; Schuller, A.; van der Sluis, L.; Tjakkes, G. Dietary inflammatory potential is associated with poor periodontal health: A population-based study. Journal of clinical periodontology 2021, 48, 907-918, doi:10.1111/jcpe.13472.
  25. Sung, C.; Huang, R.; Cheng, W.; Kao, T.; Chen, W. Association between periodontitis and cognitive impairment: Analysis of national health and nutrition examination survey (NHANES) III. Journal of clinical periodontology 2019, 46, 790-798, doi:10.1111/jcpe.13155.
  26. Eke, P.I.; Borgnakke, W.S.; Genco, R.J. Recent epidemiologic trends in periodontitis in the USA. Periodontol 2000 2020, 82, 257-267, doi:10.1111/prd.12323.
  27. Hwang, S.; Jang, J.; Park, J. Association between Healthy Lifestyle (Diet Quality, Physical Activity, Normal Body Weight) and Periodontal Diseases in Korean Adults. International journal of environmental research and public health 2022, 19, doi:10.3390/ijerph19073871.
  28. Watson, S.; Woodside, J.; Winning, L.; Wright, D.; Srinivasan, M.; McKenna, G. Associations between self-reported periodontal disease and nutrient intakes and nutrient-based dietary patterns in the UK Biobank. Journal of clinical periodontology 2022, 49, 428-438, doi:10.1111/jcpe.13604.
  29. Deng, K.; Pelekos, G.; Jin, L.; Tonetti, M. Diagnostic accuracy of self-reported measures of periodontal disease: A clinical validation study using the 2017 case definitions. Journal of clinical periodontology 2021, 48, 1037-1050, doi:10.1111/jcpe.13484.
  30. Ortigara, G.; Mário Ferreira, T.; Tatsch, K.; Romito, G.; Ardenghi, T.; Sfreddo, C.; Moreira, C. The 2018 EFP/AAP periodontitis case classification demonstrates high agreement with the 2012 CDC/AAP criteria. Journal of clinical periodontology 2021, 48, 886-895, doi:10.1111/jcpe.13462.
  31. Montero, E.; Herrera, D.; Sanz, M.; Dhir, S.; Van Dyke, T.; Sima, C. Development and validation of a predictive model for periodontitis using NHANES 2011-2012 data. Journal of clinical periodontology 2019, 46, 420-429, doi:10.1111/jcpe.13098.
  32. Austin, P. An Introduction to Propensity Score Methods for Reducing the Effects of Confounding in Observational Studies. Multivariate behavioral research 2011, 46, 399-424, doi:10.1080/00273171.2011.568786.
  33. Guo, B.; Guo, Y.; Nima, Q.; Feng, Y.; Wang, Z.; Lu, R.; Baimayangji; Ma, Y.; Zhou, J.; Xu, H.; et al. Exposure to air pollution is associated with an increased risk of metabolic dysfunction-associated fatty liver disease. Journal of hepatology 2022, 76, 518-525, doi:10.1016/j.jhep.2021.10.016.
  34. Giuliani, T.; De Pastena, M.; Paiella, S.; Marchegiani, G.; Landoni, L.; Festini, M.; Ramera, M.; Marinelli, V.; Casetti, L.; Esposito, A.; et al. Pancreatic Enucleation Patients Share the Same Quality of Life as the General Population at Long-Term Follow-Up: A Propensity-Score Matched Analysis. Annals of surgery 2021, doi:10.1097/sla.0000000000004911.
  35. Stanton, C.; Bansal-Travers, M.; Johnson, A.; Sharma, E.; Katz, L.; Ambrose, B.; Silveira, M.; Day, H.; Sargent, J.; Borek, N.; et al. Longitudinal e-Cigarette and Cigarette Use Among US Youth in the PATH Study (2013-2015). Journal of the National Cancer Institute 2019, 111, 1088-1096, doi:10.1093/jnci/djz006.
  36. Gerritsen, J.; Zwarthoed, R.; Kilgallon, J.; Nawabi, N.; Jessurun, C.; Versyck, G.; Pruijn, K.; Fisher, F.; Larivière, E.; Solie, L.; et al. Effect of awake craniotomy in glioblastoma in eloquent areas (GLIOMAP): a propensity score-matched analysis of an international, multicentre, cohort study. The Lancet. Oncology 2022, doi:10.1016/s1470-2045(22)00213-3.
  37. Lyu, N.; Wang, X.; Li, J.; Lai, J.; Chen, Q.; Li, S.; Deng, H.; He, M.; Mu, L.; Zhao, M. Arterial Chemotherapy of Oxaliplatin Plus Fluorouracil Versus Sorafenib in Advanced Hepatocellular Carcinoma: A Biomolecular Exploratory, Randomized, Phase III Trial (FOHAIC-1). Journal of clinical oncology : official journal of the American Society of Clinical Oncology 2022, 40, 468-480, doi:10.1200/jco.21.01963.
  38. Qian, D.; Kleber, T.; Brammer, B.; Xu, K.; Switchenko, J.; Janopaul-Naylor, J.; Zhong, J.; Yushak, M.; Harvey, R.; Paulos, C.; et al. Effect of immunotherapy time-of-day infusion on overall survival among patients with advanced melanoma in the USA (MEMOIR): a propensity score-matched analysis of a single-centre, longitudinal study. The Lancet. Oncology 2021, 22, 1777-1786, doi:10.1016/s1470-2045(21)00546-5.

Round 2

Reviewer 2 Report

Paper accepted for publication. Please mention any conflict of interest, if any.